# Autonomous Tracking of ShenZhou Reentry Capsules Based on Heterogeneous UAV Swarms

**Boxin Li [1], Boyang Liu [1,2], Dapeng Han [1] and Zhaokui Wang [1,*]**

1   School of Aerospace Engineering, Tsinghua University, Beijing 100084, China
2   Jiuquan Satellite Launch Centre, Lanzhou 732750, China
*   Correspondence: wangzk@mail.tsinghua.edu.cn

**Abstract:** The safe landing and rapid recovery of the reentry capsules are very important to manned spacecraft missions. A variety of uncertain factors, such as flight control accuracy and wind speed, lead to a low orbit prediction accuracy and a large landing range of reentry capsules. It is necessary to realize the autonomous tracking and continuous video observation of the reentry capsule during the low-altitude phase. Aiming at the Shenzhou return capsule landing mission, the paper proposes a new approach for the autonomous tracking of Shenzhou reentry capsules based on video detection and heterogeneous UAV swarms. A multi-scale video target detection algorithm based on deep learning is developed to recognize the reentry capsules and obtain positioning data. A self-organizing control method based on virtual potential field is proposed to realize the cooperative flight of UAV swarms. A hardware-in-the-loop simulation system is established to verify the method. The results show that the reentry capsule can be detected in four different states, and the detection accuracy rate of the capsule with parachute is 99.5%. The UAV swarm effectively achieved autonomous tracking for the Shenzhou reentry capsule based on the position obtained by video detection. This is of great significance in the real-time searching of reentry capsules and the guaranteeing of astronauts' safety.

**Keywords:** reentry capsules; autonomous tracking; UAV swarm flight control; video detection

## 1. Introduction

Human spaceflight is becoming more frequent. There are already 13 manned spaceflight missions in 2021, and this number will continue to increase in 2022. The rapid search and rescue of the reentry capsules is very important the manned spacecraft missions [1,2]. China's manned spaceflight project is in the stage of orbital construction of a space station. The construction of the China's space station will be completed in 2022, and there will be more frequent spacecraft reentries to Earth. Spacecraft landing search and rescue missions will be normalized. It is necessary to observe and track the reentry capsules during the reentry process to guarantee real-time search and rescue.

The tracking and observation of the reentry capsules usually relies on tracking, telemetry, and command (TT&C) communications and radar detection [3]. The Apollo program manned spacecraft used multiple ground stations to locate and track the reentry capsules. The location of the reentry capsule was obtained through data processing on the ground [4,5]. Another solution used in the tracking and location of manned spacecraft's reentry capsules is radar detection. For the location of reentry capsules, the Japan Aerospace Exploration Agency (JAXA) set up the ground optical system (GOS), the directional finding system (DFS), and the marine radar system (MRS). In the Hayabusa2 reentry capsule recovery operation, marine radars were used to estimate the landing point of the capsule. The tracking was accurate up to 240 m from the landing point [6–8]. The Shenzhou series of manned spacecraft missions used radar detection to track reentry capsules before they entered the black-out area and to and predict the location of the landing site. After escaping the black-out area, the unified S-band measurement device was used to track the reentry

capsule until the main parachute deployed. The rescue team carried out search and rescue in the predicted landing area according to the location of the parachute deployed. However, the communication between the reentry capsule and the ground stations is interrupted in the black-out area. The real-time status of the reentry capsule cannot be known. Various uncertain factors, such as flight control accuracy and wind speed, may result in an inaccurate landing prediction and a large landing range of reentry capsules. Especially in the low-altitude parachute stage, the ground tracking devices influenced by the block of ground objects are difficult to operate normally. The radars have poor measurement coverage below 10 km altitude. In order to improve the search and rescue speed of the reentry capsule and ensure the safety of astronauts, it is necessary to research real-time autonomous tracking and video detection during the reentry capsule landing process. A low-altitude measurement method with maneuvering capabilities should be developed.

Recently, unmanned aerial vehicle (UAV) swarms have been widely used for mobile target searching and autonomous tracking [9]. UAV swarms can cover larger areas and improve the speed of the search and tracking missions because of their excellent mobility and collaboration. They perform well in hazardous and harsh environments in which conventional equipment cannot work or humans cannot stay [10]. The Wide Area Search Munitions (WASM) project established the multi-UAV collaborative control simulation platform and used layered control and optimization technology to enhance the coordinated global search capability of UAV swarms under the background of complex tasks [11]. The America Office of Naval Research announced the Low-Cost UAV Swarming Technology (LOCUST) program in 2015. The deployment of UAV swarms will reduce hazards and free up personnel to perform more complex tasks as well as requiring fewer people to conduct multiple missions [12]. With the increasing applications of UAV swarms, a growing number of UAV swarm control algorithms have been studied to achieve mobile target tracking and searching.

Rezgui, J. et al. [13] propose a cooperative UAVs framework named CF-UAVs-MTST for simulating mobile target search and mobile target tracking algorithms and approaches. It can simulate UAV software and hardware structure, but it does not support 3D target motion estimation and does not consider cases in which the target speed is considerably higher than the UAV flight speed. In [14–16], approaches of autonomous tracking for UAV swarms were proposed to localize the radio frequency (RF) mobile targets based on received signal strength (RSS) measurements. Researchers in [14,17,18] proposed multi-agent-reinforcement-learning-based methods to improve the UAV swarm's performance and generalization. Long biological evolution and natural selection have created the amazing phenomenon of swarm intelligence in nature, which inspires some new solutions to deal with complex swarm control problems [19–21]. Researchers are inspired by the swarm intelligence of wolves to solve concerned problems in cooperative control technology in [22,23]. Among the most bio-inspired cooperative approaches [24,25], the swarm intelligence of wolves is used make task assignment more reasonable and improve efficiency, rather than focusing on UAV swarms' cooperative control.

Most of the related work introduced above only considered two-dimensional UAV path planning and depends on the radio frequency signal. The bio-inspired cooperative approaches to UAV swarms focus more on reasonable task assignment. For the descent and landing process of the reentry capsules, it is necessary to study a unique autonomous tracking method that can deal with changes in different altitudes and positions. Meanwhile, in the low-altitude stage, the method for reentry capsule recognition and detection based on vision should be studied to obtain continuous images and real-time status of the reentry capsules. Visual detection based on deep learning has been widely used in robots, UAVs, and other intelligent agents. It enhances the performance of target detection and tracking.

Aiming at the challenge of Shenzhou reentry capsule tracking, this paper proposes a new approach for autonomous tracking based on video detection and heterogeneous UAV swarms. A scheme for mission assignment and autonomous tracking strategy is designed to satisfy the different states of reentry capsules. A multi-scale video target detection

algorithm based on deep learning is developed to recognize Shenzhou reentry capsules and obtain positioning data. Additionally, a self-organizing control method based on virtual potential field for the cooperative flight of UAV swarms is studied. In order to verify the performance of the proposed approach, we establish a hardware-in-the-loop simulation system and a novel reentry capsules dataset.

The paper is organized as follows: Section 2 provides the overall scheme of the autonomous tracking strategy for reentry capsules. In Section 3, the intelligent video detection algorithm for Shenzhou reentry capsules is described. In Section 4, the cooperative flight control method for UAV swarms is provided. Section 5 introduces the hardware-in-the-loop simulation system and analyses the test results. Discussions and conclusions are presented in Section 6.

## 2. Autonomous Tracking Strategy

The wide range of landing sites of the reentry capsules is a challenge for search and rescue. The theoretical range can be hundreds of square kilometers. For safety reasons, ground communication equipment and rescue teams are usually deployed in safe areas far from the landing site. The search and rescue teams will not go to rescue until the reentry capsule has landed. Moreover, due to the influence of the curvature of the earth, topography and disturbance of ground clutters, both optical equipment and radar may be invalid when measuring the pitch angle. UAV swarms with vision payloads do not have blind zones for measurement and can make use of their excellent flexibility and mobility to improve observation conditions and coverage of the landing area. This paper proposes an autonomous tracking system based on video detection and heterogeneous UAV swarms to realize the low-cost and fast tracking of Shenzhou reentry capsules.

### 2.1. Overview

The autonomous tracking system for Shenzhou reentry capsules contains heterogeneous UAV swarms, electro-optical pods and the video detection module, communication modules, and ground stations, as shown in Figure 1. The heterogeneous UAV swarms consist of fixed-wing UAVs and rotary-wing UAVs. The fixed-wing UAVs are responsible for the high-altitude tracking of the capsules' reentry process owing to their characteristics of long flight distance and high flight speed. They are the central nodes of the swarm communication and the mission planning centers that ensure the transmission of data and images between the swarms and the ground station. The rotary-wing UAVs have the characteristics of excellent flexibility and stable hovering ability. They are used for low-altitude tracking of the capsules' reentry process and high-quality image capture.

The kinematic parameters of the two types UAVs are determined by investigating the existing UAV models in Figure 2. For the fixed-wing UAVs, taking the Twin-Tailed Scorpion as an example, the flight ceiling is usually around 10 km and the flight speed is about 300 km/h. For the rotary-wing UAVs, the flight ceiling is usually around 5 km and the flight speed is about 80 km/h. A typical model is X-Swift, made by AOSSCI company. We have only selected two specific types whose parameters are applicable to our task. Moreover, other types with similar parameters can also be used. The kinematic parameters of the heterogeneous UAV swarms are shown in Table 1.

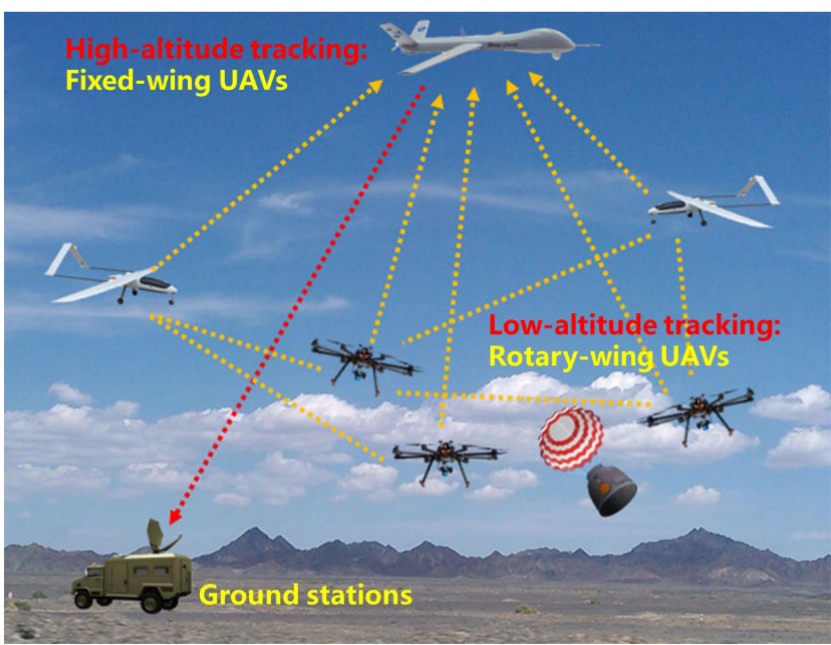

**Figure 1.** The autonomous tracking system for Shenzhou reentry capsules.

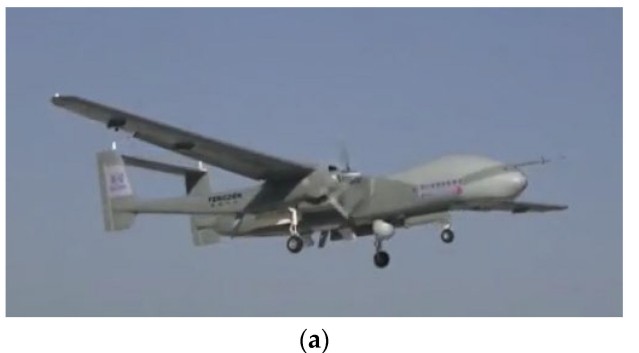

(**a**)

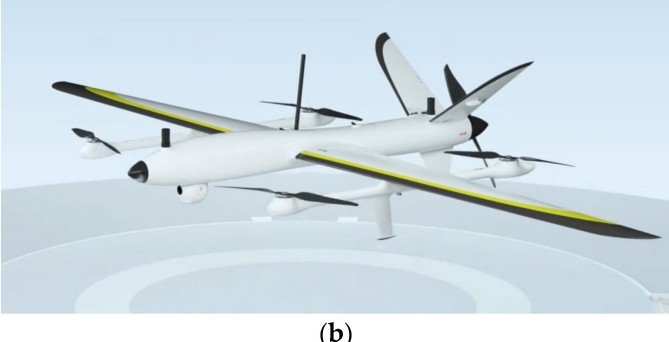

(**b**)

**Figure 2.** The heterogeneous UAVs. (**a**) Twin-Tailed Scorpion. (**b**) X-Swift.

**Table 1.** Kinematic parameters of the heterogeneous UAV swarms.

| Parameters | Fixed-Wing UAVs | Rotary-Wing UAVs |
| --- | --- | --- |
| Type | Twin-Tailed Scorpion | X-Swift |
| Service Ceiling | 10 km | 6 km |
| Maximum speed | 300 km/h | 120 km/h |
| Maximum endurance | 35 h | 6 h |
| Maximum Range | 6000 km | 100 km |
| MTOW | 2800 kg | 25 kg |
| Payload | Electro-optical pod Maximum image distance 100 km | Electro-optical pod Maximum image distance 1 km |

*2.2. Autonomous Tracking Strategy Design*

Due to the huge range of speed changes and the wide distribution of UAV swarms during the landing process of the reentry capsule, a single control mode cannot support continuous and efficient control of heterogeneous UAV swarms throughout the whole reentry process.

In order to satisfy the special requirements of the Shenzhou reentry capsule tracking and improve the mission effectiveness, the autonomous tracking strategy is divided into three modes, as shown in Figure 3. The three modes are initial configuration, relay

tracking and observation, and aggregation. Each mode is suitable for different mission scenarios. The transition between modes depends on the status of the reentry capsule's landing process.

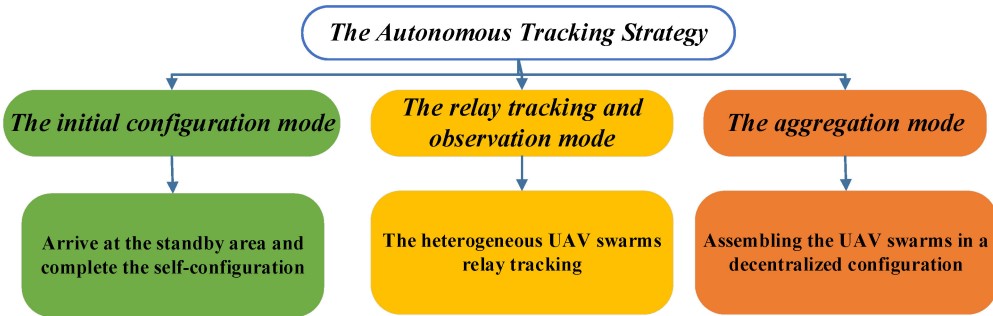

**Figure 3.** The overall autonomous tracking strategy.

The initial configuration mode is applied before the capsule reentry. In this mode, the fixed-wing UAVs fly to the standby area according to the predicted reentry trajectory, and the low-altitude rotary-wing UAVs go to the predicted landing area and complete self-configuration. A relatively accurate orbit forecast can be given based on the last three orbits before the spacecraft reentry, so the UAV swarms should start the initial configuration mode within 3 to 4 h before the capsule reentry. The rotary-wing UAVs should complete self-configuration around the predicted landing site, and the fixed-wing UAVs should arrive at the standby area before the capsule reentry. In order to ensure the complete coverage of the reentry capsule in the low-altitude stage, the number and distribution density of the rotary-wing UAVs are designed by the effective range of the onboard measurement equipment.

The relay tracking and observation mode is applied after capsule reentry. Because the horizontal velocity of the reentry capsule is much larger than the UAV's speed, relay tracking among the swarm UAVs is required. The goal of this mode is to allow each UAV to obtain the longest possible tracking distance. The motion state of the UAV swarms will change according to the real-time position/speed of the reentry capsule, the maneuverability of the UAV, and the measurement payload constraints.

After the reentry capsule has landed and the mission has been completed, the aggregation mode is applied. The goal of this mode is to assemble the UAV swarms in a decentralized configuration.

The tracking mode will be converted with the change of mission status. The specific conversion relationship is shown in Figure 4.

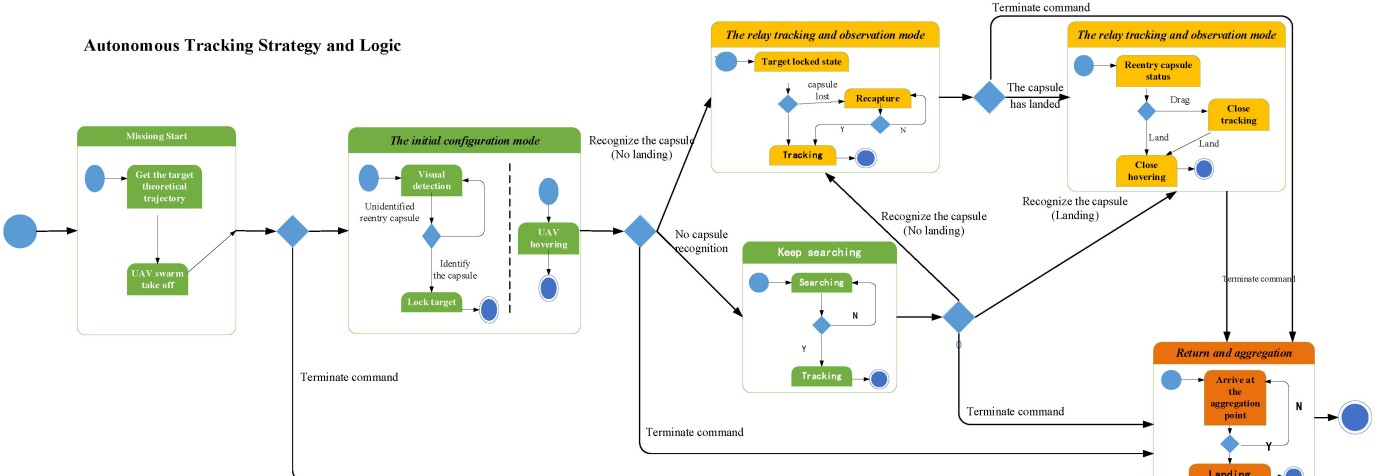

**Figure 4.** The autonomous tracking mode conversion relationship.

## 3. Video Detection for Reentry Capsules

The landing process of the reentry capsule has a certain degree of randomness due to the influence of various uncertain factors such as flight control accuracy and wind speed. The measurement scheme of the ground fixed station cannot realize the whole process observation and tracking of the reentry capsule. In particular, the ground tracking equipment will be invalid during the low-altitude landing phase because of the interference of ground obstructions. Therefore, it is necessary to realize the real-time video detection of the reentry capsule.

For the real-time detection and tracking task of the reentry capsules, a reliable target video detection algorithm for the reentry capsules should be studied. The video detection algorithm should achieve accurate and efficient detection of the reentry capsules under complex backgrounds with various weather conditions and different distances. Moreover, the scale of the return capsule is constantly changing during the descent. There is a large difference between the state of the single capsule and the state of the capsule with umbrella. In order to solve the above problem, a multi-scale video target detection algorithm based on deep learning is developed to recognize the reentry capsules and obtain positioning data.

### 3.1. Target Detection Network for the Reentry Capsule

The autonomous tracking mission of Shenzhou Reentry Capsules has high requirements for the acquisition success rate, continuous tracking stability, real-time performance, and search efficiency. Existing object detection methods are mostly categorized by whether they have a region-of-interest proposal step (two-stage) or not (one-stage) [26]. While two-stage detectors tend to be more flexible and more accurate, one-stage detectors are often considered to be simpler and more efficient by leveraging predefined anchors [27]. YOLO, an acronym for "You only look once", is a typical one-stage detection algorithm which has been widely used in real-time object detection tasks. It completes the prediction of the classification and location information of the objects according to the calculation of the loss function, so it makes the target detection problem transform into a regression problem solution [28]. YOLOv5 has the best performance among YOLO algorithms. It can balance detection accuracy and model complexity under the constraints of processing platforms with limited memory and computation resources [29].

Aiming at the problem of autonomous Shenzhou reentry capsule tracking, our paper used YOLOv5 based on regression method to realize the video detection of reentry capsules video detection. The architecture of the network consists of for parts: (a) input, (b) backbone, (c) neck, (d) output [30]. Localization and classification of different scales can be achieved. The algorithm can not only ensure the detection speed but can also ensure detection accuracy. The network structure is shown in the following Figure 5.

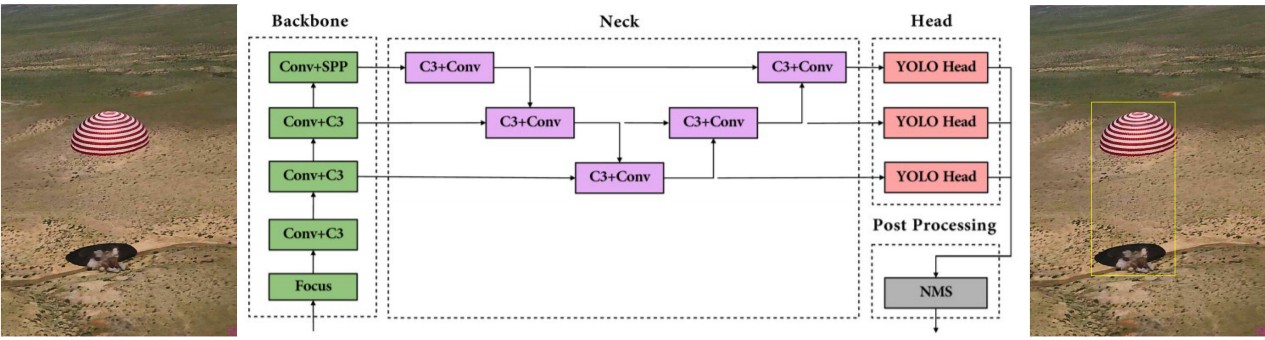

**Figure 5.** The network structure.

### 3.2. Reentry Capsules Dataset

For the video detection of the reentry capsules during the landing process, a dataset named DSSlcapsule is constructed to train and test the network. We use visible light image data as the image source of the dataset. The DSSLcapsule dataset contains images

of the reentry capsules under different weather conditions, different viewing angles, and partial occlusion, as shown in Figure 6. Most of the data come from the videos and images of Shenzhou manned spaceflight missions. In addition, we collected some open-source videos to expand the dataset. It is beneficial to train a more robust detection network by establishing a richer image dataset.

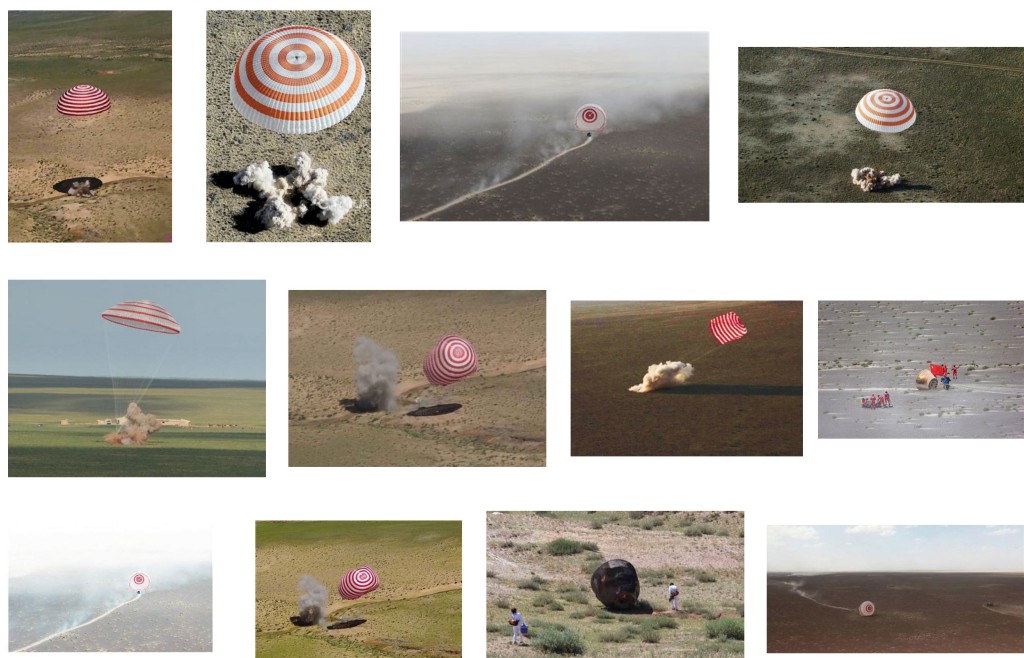

**Figure 6.** DSSlcapsule dataset.

　　The image data containing the reentry capsules should be labelled. The labeling information includes the real location bounding box and classification category. Image data without the labeling, such as the environment, will be used as negative samples during network training. The LabelTool program written in Python language under Ubuntu is used to label the landing target position and category label in the image, as shown in Figure 7.

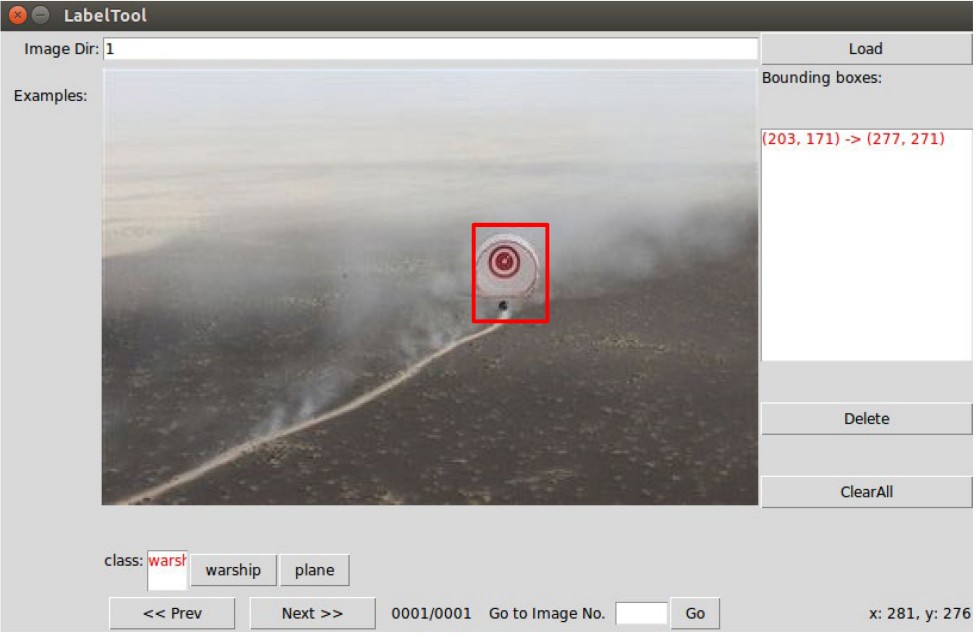

**Figure 7.** Dataset labeling.

### 3.3. Network Training and Validation

The dataset is randomly divided into a training sample set and a validation sample set. The training sample set accounts for 70%, and the validation sample set accounts for 30%. The network is trained on an NVIDIA TITAN X using stochastic gradient descent.

A parallel computing server hardware environment based on multiple high-performance GPU parallel computing units is constructed. High-performance parallel computing capabilities are realized, and faster deep learning object detection algorithms are supported by bridging multiple GPU parallel computing units. The NVIDIA TESLA K80-type GPU parallel computing unit was selected to accelerate the operation of more than 400 high-performance computing applications and all major deep learning frameworks, as shown in Figure 8.

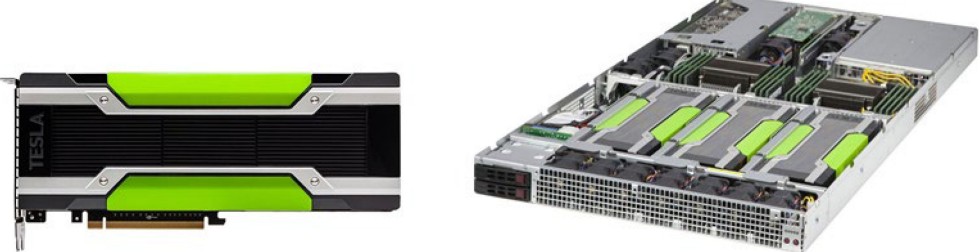

**Figure 8.** High-performance parallel computing server based on TESLA K80 parallel computing unit.

The network training parameters are set as shown in Table 2, including parameters such as learning rate of network training, learning rate decay mode, maximum number of training iterations, weight decay factor, and operation mode.

**Table 2.** Network training parameters.

| Parameters | Value |
| --- | --- |
| Initial learning rate | 0.001 |
| Learning rate decay constant | 0.1 |
| Total number of iterations | 80,000 |
| Weight decay factor | 0.0005 |
| Optimizer type | SGD |
| Calculation method of AP | 11-point |

The training dataset images contain location bounding boxes and class labels. The loss function is defined as the sum of the deviations of the object localization and classification confidence based on the regression method.

$$L(x, c, l, g) = \frac{1}{N}(L_{\text{conf}}(x, c) + \alpha L_{\text{loc}}(x, l, g)) \tag{1}$$

$N$ is the number of default bounding boxes that match the real bounding box. $L_{\text{loc}}(x, l, g)$ is the position deviation, obtained using the Smooth L1 LOSS method. $L_{\text{conf}}(x, c)$ is the classification deviation, obtained by the Softmax LOSS method. the input of $L_{\text{conf}}(x, c)$ is the confidence of each category, and the weight item $\alpha$ is set to 1. Through the iterative training of a large number of sample data according to the above method, the continuous iterative optimization of the entire network parameters is realized. The optimal network model parameters can be obtained. The training results are shown in Figure 9. Real image data are used for extended training based on the training with simulated capsule data. The training speed is fast with high accuracy, and the loss convergence effect is good. While the simulated dataset is relatively small, there is a slight overfitting.

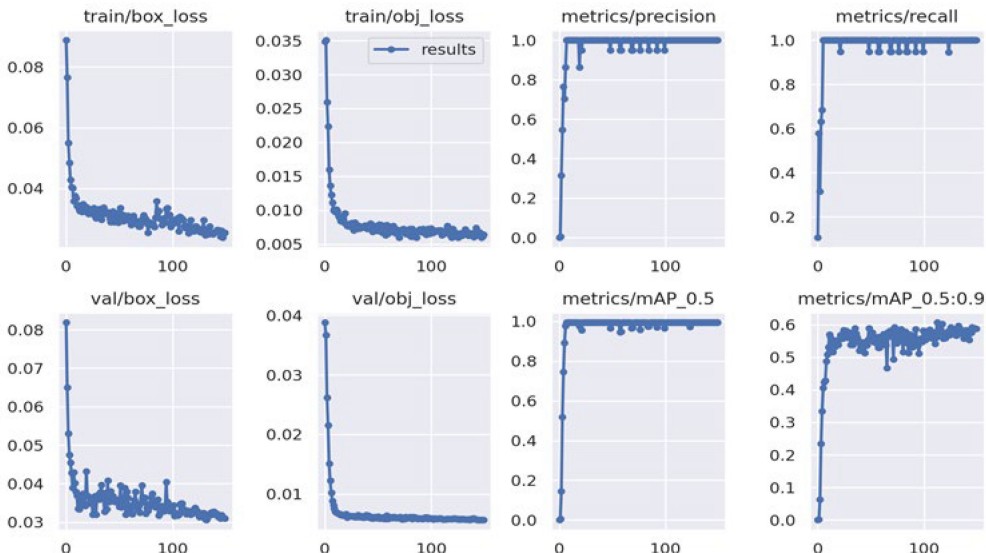

**Figure 9.** Training results.

The weight parameters obtained by training iterations are used for intelligent identification and detection of the Shenzhou reentry capsules. The images containing the reentry capsule in the validation dataset were tested, and the results are shown in Figure 10. All the reentry capsule targets were all detected.

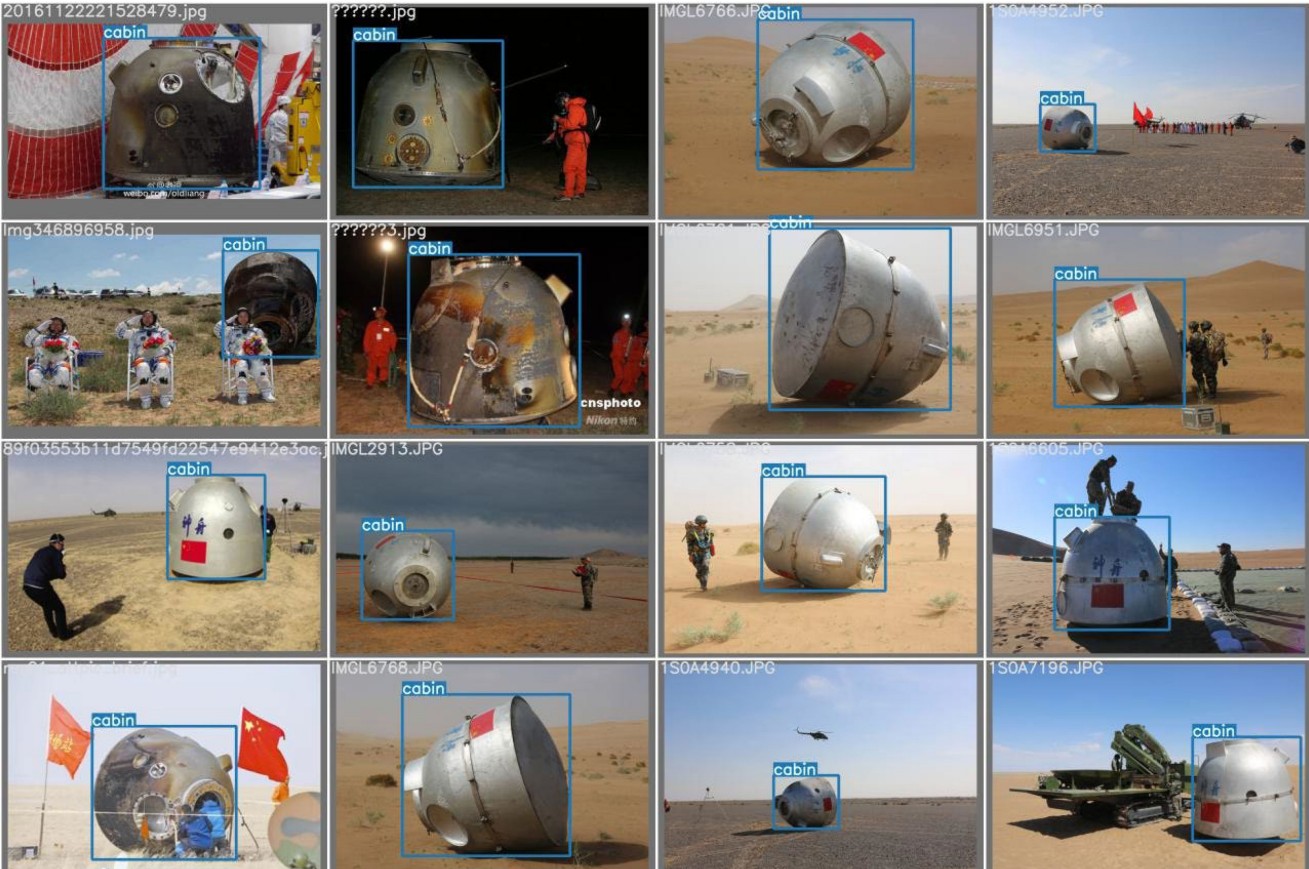

**Figure 10.** Shenzhou reentry capsule detection test.

### 4. Cooperative Flight Control Method of UAV Swarms

According to the real-time results of the return capsule detection from the videos, the heterogeneous UAV swarms will perform the autonomous tracking task of the reentry capsule. The UAV swarms will carry out autonomous tracking according to the three strategy modes designed in Section 2.2. The three strategy modes are initial configuration, relay tracking and observation, and aggregation. Each mode is suitable for different mission scenarios.

For the cooperative flight control of UAV swarms, we proposed a self-organizing control based on virtual potential field. Each type of strategy modes is combined proportionally with virtual potential force according to the control expectation in order to achieve the continuous tracking of the reentry capsule.

*4.1. Virtual Potential Field Design*

The coordinate system of the models in this paper is the geocentric fixed system. The position vector of the target is expressed as $X = [x, y, z]^{\mathrm{T}}$. The velocity vector is expressed as $v = [v_{\mathrm{x}}, v_{\mathrm{y}}, v_{\mathrm{z}}]^{\mathrm{T}}$. The distance dimension is unified as km, and the time dimension is unified as s.

4.1.1. UAV Swarms Configuration Potential Field

The purpose of the swarm configuration potential field is to distribute the UAVs in the standby area autonomously and evenly. The force of the configuration potential field controls the distance among the UAVs within a certain range.

The force on the UAVs under the configuration potential field can be expressed as

$$F_{\mathrm{sc}}(X_{\mathrm{i}}) = \frac{1}{n} \sum_{\mathrm{j} \neq \mathrm{i}}^{n} F_{\mathrm{c}}(X_{\mathrm{i}}, X_{\mathrm{j}}) \tag{2}$$

where the $n$ is the sum of all UAVs that exert virtual force. $X_{\mathrm{i}}$ and $X_{\mathrm{j}}$ are the coordinate positions of the UAV i and UAV j, respectively. The $F_{\mathrm{c}}(X_{\mathrm{i}}, X_{\mathrm{j}})$ is the virtual forces from the UAV j on the UAV i, which can be expressed as Equation (3).

$$F_{\mathrm{c}}(X_{\mathrm{i}}, X_{\mathrm{j}}, R_c, R_e) = -k_{\mathrm{c}}(X_{\mathrm{i}}, X_{\mathrm{j}}, R_c, R_e) \cdot F_{\max} \frac{X_{\mathrm{i}} - X_{\mathrm{j}}}{\|X_{\mathrm{i}} - X_{\mathrm{j}}\|} \tag{3}$$

where the $F_{\max}$ is the maximum maneuvering force that the UAV power system can provide. The $F_{\mathrm{c}}$ is the repulsive force when the distance between the UAVs is smaller than the desired distance, and it is the attractive force when the distance between the UAVs is greater than the desired distance. There is always a trend of movement towards the desired distance between UAVs. Therefore, the coefficient function $k_{\mathrm{c}}(X_{\mathrm{i}}, X_{\mathrm{j}})$ is designed as Equation (4),

$$k_{\mathrm{c}}(X_{\mathrm{i}}, X_{\mathrm{j}}, R_c, R_e) = \begin{cases} 0 & R_{\mathrm{e}} \leq \|X_{\mathrm{i}} - X_{\mathrm{j}}\| \\ \frac{C_{\mathrm{a}}}{\|X_{\mathrm{i}} - X_{\mathrm{j}}\|^2} & R_{\mathrm{c}} < \|X_{\mathrm{i}} - X_{\mathrm{j}}\| \leq R_{\mathrm{e}} \\ \frac{-C_{\mathrm{a}}}{\|X_{\mathrm{i}} - X_{\mathrm{j}}\|^2} & \sqrt{C_{\mathrm{a}}} < \|X_{\mathrm{i}} - X_{\mathrm{j}}\| \leq R_{\mathrm{c}} \\ -1 & \|X_{\mathrm{i}} - X_{\mathrm{j}}\| \leq \sqrt{C_{\mathrm{a}}} \end{cases} \tag{4}$$

where the $R_{\mathrm{c}}$ is the desired distance, $C_{\mathrm{a}}$ is the constant that determines the magnitude of $F_{\mathrm{c}}$, and $R_{\mathrm{e}}$ is the maximum distance of $F_{\mathrm{c}}$. Since the ideal configuration result is a spatial structure composed of equilateral triangles, each UAV node can only have an effect on other adjacent UAV nodes when $R_{\mathrm{e}} < \sqrt{3} \cdot R_{\mathrm{c}}$ and $C_{\mathrm{a}} \leq \frac{R_c^2}{2\sqrt{3}}$. Moreover, the swarm configuration result will not be affected by the scale of swarm. To facilitate practical application, $R_{\mathrm{e}}$ can be set to $1.5R_{\mathrm{c}}$ and the $C_{\mathrm{a}}$ can be set to $\frac{R_c^2}{3.6}$.

### 4.1.2. Safety Potential Field

The safety potential field is to enable the UAV to avoid the dangerous area. Under the safety potential field, the distance between the controlled UAV and the dangerous point can be greater than the safe distance $R_s$. The safety potential field is designed as Equation (5),

$$U_s(r, R_s, C_s) = \begin{cases} 0 & R_s < r \\ \frac{C_s}{r} - \frac{C_s}{R_s} & \sqrt{C_s} < r \le R_s \\ 2\sqrt{C_a} - \frac{C_s}{R_s} - r & r \le \sqrt{C_s} \end{cases} \tag{5}$$

Therefore, the safety control force $F_s$ is expressed as

$$\begin{aligned} F_s(X_i, X_d, R_s, C_s) &= -\Delta U_s(\|X_i - X_d\|, R_s, C_s) F_{max} \\ &= -k_s(X_i, X_d, R_s, C_s) \frac{(X_i - X_T)}{\|X_i - X_T\|} F_{max} \end{aligned} \tag{6}$$

where $X_d$ is the coordinate of the dangerous point and $k_s$ is the coefficient function of the safety control force. $k_s$ can be expressed as (7),

$$k_s(X_i, X_d, R_s, C_s) = \begin{cases} 0 & R_s \le \|X_i - X_d\| \\ \frac{-C_s}{\|X_i - X_d\|^2} & \sqrt{C_s} < \|X_i - X_d\| \le R_s \\ -1 & \|X_i - X_d\| \le \sqrt{C_s} \end{cases} \tag{7}$$

### 4.1.3. Central Gravitational Potential Field

The autonomous tracking of the reentry capsule is the core of the mission. It is the basis that the reentry capsule is within the measurement range of the UAV's onboard measurement equipment. Therefore, a central gravitational potential field is designed to make the UAV move toward the reentry capsule and keep a safe distance between the UAV and the reentry capsule. The central gravitational potential field is designed as Equation (8),

$$U_g(r, R_s, R_e, C_g) = \begin{cases} C_g \ln(R_e) - C_g \ln(R_s) & R_e \le r \\ C_g \ln(r) - C_g \ln(R_s) & R_s \le r < R_e \\ C_g \ln(R_s) - C_g \ln(r) & C_g \le r < R_s \\ C_g \ln(R_s) - C_g \ln(C_a) + C_g - r & r < C_g \end{cases} \tag{8}$$

where $R_s$ is the desired safety distance, $R_e$ is the force distance, and $C_g$ is adjusted as a proportional coefficient constant according to the actual situation. The central gravitational force $F_g$ of the spacecraft at $X_T$ point on the UAV at $X_i$ point can be expressed as Equation (9).

$$\begin{aligned} F_g(X_i, X_T, R_s, R_e, C_g) &= -\Delta U_g(\|X_i - X_T\|, R_s, R_e, C_g) F_{max} \\ &= -k_g(X_i, X_T, R_s, R_e, C_g) \frac{(X_i - X_T)}{\|X_i - X_T\|} F_{max} \end{aligned} \tag{9}$$

$$k_g(X_i, X_T, R_s, R_e, C_g) = \begin{cases} 0 & R_e \le \|X_i - X_T\| \\ \frac{C_g}{\|X_i - X_T\|} & R_c \le \|X_i - X_T\| < R_e \\ -\frac{C_g}{\|X_i - X_T\|} & C_g \le \|X_i - X_T\| < R_c \\ -1 & \|X_i - X_T\| < C_g \end{cases} \tag{10}$$

### 4.1.4. Altitude Potential Field

If only the central gravitational potential field is used, the UAV will track the top of the reentry capsule during the descent of the reentry capsule and the ideal observation angle cannot be obtained. Therefore, an altitude potential field is required to make UAVs accom-

pany reentry capsule at the same altitude. Similar to the design of the central gravitational potential field, the altitude potential field $U_H(X_i, H_T)$ is expressed as Equation (11),

$$U_H(X_i, H_T) = \begin{cases} \|X_i\| - H_T - 6371 & \|X_i\| > H_T + 6371 \\ H_T + 6371 - \|X_i\| & \|X_i\| \leq H_T + 6371 \end{cases} \tag{11}$$

The altitude control force of the UAV can be expressed as (11),

$$F_H(X_i, H_T) = -\Delta U_H(X_i, H_T) F_{max} = -k_H(X_i, H_T) \cdot \frac{X_i}{\|X_i\|} \cdot F_{max} \tag{12}$$

where $H_T$ is the altitude of the reentry capsule. $k_H(X_i, H_T)$ is the altitude control force coefficient, and its expression is Equation (13).

$$k_H(X_i, H_T) = \begin{cases} 1 & \|X_i\| > H_T + 6371 \\ -1 & \|X_i\| \leq H_T + 6371 \end{cases} \tag{13}$$

The force of the altitude potential field is relatively strong, and has a strong constraint on the movement of the UAV.

### 4.1.5. Virtual Wind Potential Field

Inspired by the air resistance in the real physical environment, a virtual damping force is designed to dampen the vibration. The virtual air damping is designed as (14),

$$F_v(v_i, v_{vw}) = k_v(v_i, v_{vw}) \frac{v_i - v_{vw}}{\|v_i - v_{vw}\|} \cdot F_{max} \tag{14}$$

$$k_v(v_i, v_{vw}) = -\frac{1}{2} C_v \|v_i - v_{vw}\|^2 \tag{15}$$

where $v_i$ is the velocity vector of the controlled UAV and $v_{vw}$ is the velocity vector of the virtual wind. $C_v$ is the proportional coefficient used to adjust the size of the $F_v(v_i)$. According to the actual situation, the maximum change of the flight speed is $2v_{max}$. At this time, the max force that the virtual wind can provide is $F_{max}$, so the expression of $C_v$ is as follows

$$C_v = 2\frac{1}{(2v_{max})^2} \tag{16}$$

Correspondingly, $k_v(v_i, v_{vw})$ is expressed as

$$k_v(v_i, v_{vw}) = \begin{cases} \frac{1}{4v_{max}^2} \|v_i - v_{vw}\|^2 & \|v_i - v_{vw}\| < 2v_{max} \\ 1 & \|v_i - v_{vw}\| \geq 2v_{max} \end{cases} \tag{17}$$

The virtual wind force is a non-conservative force, which makes the speed of the UAV tend to the speed of the wind field.

### 4.2. Heterogeneous UAV Swarm Control Based on Virtual Potential Field

According to the UAV swarms cooperative flight modes and the real state of the reentry capsule, the proportional combinations of virtual potential forces are divided into three categories to realize heterogeneous UAV swarm cooperative flight control.

### 4.2.1. Initial Configuration and Cruise

The goal of the initial configuration and cruise is to make the UAV swarms arrive at the standby area and distribute in the predicted landing area. In this strategy mode, the flight control force is composed of the configuration force, safety control force, altitude control force, and virtual wind force. The expression is (18)

$$F_d(X_i) = \frac{1}{3}\left( F_{sc}(X_i, R_c, R_e) + \frac{1}{m}\sum_{d\neq i}^{m} F_s(X_i, X_d, R_d, C_s) \right.$$
$$\left. + k_{fH}F_H(X_i, H_T) + k_{fv}F_v(v_i, v_T) \right) \tag{18}$$

where $F_{sc}$ control the UAVs to complete the swarm self-configuration. $m$ represents the number of danger points and the fixed-wing UAVs. $F_s$ is responsible for controlling the UAV to avoid the dangerous area. $F_H$ is used for keeping the UAV at the desired altitude. $k_{fv}F_v(v_i, v_T)$ is used to stabilize the flight speed of the UAV. $v_T$ is virtual wind speed, designed as the component of $v_i$ in the vertical direction. This design method can reduce the maneuvering vibration of the UAV in the vertical direction without affecting the maneuvering in the horizontal direction. The expression (19),

$$v_T = v_i - \frac{v_i \cdot X_i}{\|X_i\|} \cdot \frac{X_i}{\|X_i\|} \tag{19}$$

$k_{fv}$ and $k_{fH}$ are the speed control coefficient and the altitude control coefficient respectively. They are designed as follows:

$$k_{fv} = \left( \frac{C_{fv}}{|\|X_i\| - 6371 - H_T| + C_{fv}} \right)^2 \tag{20}$$

$$k_{fH} = 1 - k_{fv} \tag{21}$$

It can be seen from the formula that when the UAV is closer to the desired cruising altitude, the virtual wind will play a larger role and vice versa. $C_{fv}$ is the proportional adjustment factor. The value of $C_{fv}$ is adjusted through experiments so that the UAV can quickly reach the desired cruising height under the altitude control potential field. Moreover, it will not oscillate at the cruising height.

4.2.2. Relay Tracking and Observation

In the relay tracking and observation strategy mode, the virtual control force is designed to make the UAV swarm fly to the relay tracking area as quickly as possible. When the spacecraft enters the measurement range, each UAV will realize the accompanying flight and relay tracking.

The virtual control force of the UAV is designed as Equation (22),

$$F_d(X_i) = \frac{1}{3+k_i}\left( F_{sc}(X_i, R_{si}, R_e) + \frac{1}{m}\sum_{d\neq i}^{m} F_s(X_i, X_d, R_d, C_s) \right.$$
$$+ k_i k_{fH}F_H(X_i, H_e) + k_i k_{fv}F_v(v_i, v_e)$$
$$\left. + k_i k_{rv}F_v(X_i, v_{a\_e}) + k_i k_{rg}F_g\left(X_i, X_{sc\_e}, R_s, R_{dt}, C_g\right) \right) \tag{22}$$

$k_i$ is the flag of tracking. If the reentry capsule was captured in real-time video detection, $k_i = 1$. Otherwise, $k_i = 0$. $F_{sc}(X_i, R_{si}, R_e)$ is the self-configuration control force. $R_{si}$ is the safety distance. When $k_i$ takes a value of 1, $R_{si}$ is equal to the minimum safety distance $R_{sMin}$. When $k_i$ takes a value of 0, $R_{si}$ is equal to the self-configuration distance $R_c$ in the initial configuration and cruise mode.

The function of $F_{sc}(X_i, R_{si}, R_e)$ is to keep a safe distance between the UAVs when they are tracking the capsule and to restore the initial configuration mode when they stop tracking. $F_s$ is responsible for controlling the UAV to avoid the dangerous area. $k_{fH}$ and $k_{fv}$ are a set of proportional coefficients to control the flight height stably. The expressions are the same as above.

$k_i k_{fH} F_H(X_i, H_e)$ is the height control force. $H_e$ is the desired height, and its formula is defined as follows:

$$H_e = \begin{cases} H_w & \|X_{sc}\| - 6371 \geq H_w \\ \|X_{sc}\| - 6371 & H_T \leq \|X_{sc}\| - 6371 < H_w \\ H_T & \|X_{sc}\| - 6371 < H_T \end{cases} \tag{23}$$

where $H_w$ is the expected standby height, and $H_T$ is the self-configuration height.

$k_i k_{fv} F_v(v_i, v_e)$ is the velocity control force, and $v_e$ is the component of $v_i$ in the horizontal direction.

$k_i k_{rv} F_v$, $k_i k_{rg} F_g$ are the fly-around control force and the tracking flight control force respectively. $k_{rv}$ and $k_{rg}$ are a set of control coefficients, defined as follows:

$$k_{rv} = \left( \frac{R_s}{\|X_i - X_{sc\_e}\| + R_s} \right)^2 \tag{24}$$

$$k_{rg} = 1 - k_{rv} \tag{25}$$

The functions of a and b cooperative control are as follows: (1) When the distance $X_{sc\_e}$ between the UAV $X_i$ and the target capsule is too close or too far, the UAV can be controlled to reach the desired distance $R_s$ as quickly as possible with a radial direction maneuvering route. (2) The closer $X_{sc\_e}$ is to $R_s$, the stronger the fly-around control force is.

$v_{a\_e}$ is the fly-around speed, which is defined as Equation (26).

$$v_{a\_e} = v_{sc} + v_{around} \frac{X_{sc} \times X_i}{\|X_{sc} \times X_i\|} \tag{26}$$

$v_{around}$ is the fly-around speed of the UAV. The purpose of flying around is to obtain a 360° view of the reentry capsule.

### 4.2.3. Aggregation

In the aggregation mode, the UAV swarms will fly to the assembly and recovery location. The force on each UAV is a combination of the UAV swarm configuration force, safety control force, altitude control force, virtual wind field force, and gravitational center force. The virtual control force in the aggregation mode is designed as Equation (27),

$$\begin{aligned} F_d(X_i) = \frac{1}{4} \Bigg( & F_{sc}(X_i, R_{sMin}, R_{sMin}) + \frac{1}{m} \sum_{d \neq i}^{m} F_s(X_i, X_d, R_d, C_s) \\ & + k_{fH} F_H(X_i, H_T) + k_{fv} F_v(v_i, v_T) \\ & + F_g\left(X_i, X_{ap}, R_{sap}, \infty, C_g\right) \Bigg) \end{aligned} \tag{27}$$

$F_{sc}(X_i, R_{sMin}, R_{sMin})$ is the self-configuration force of the swarms. $R_{sMin}$ is the minimum safe distance. $F_s$, $F_H$, $k_{fv} F_v$ are the same as above, and their function is to keep the UAV flying at the self-configuration altitude. The function of $F_g$ is to control the UAV to fly to the assembly and recovery area. $X_{ap}$ is the coordinate of the assembly location. $R_{sap}$ is the safety radius of the area assembly center. $C_g$ is adjusted according to the actual situation, which ensures that the UAV can receive a strong gravitational potential field even when the UAV is far away from the assembly area. It generally should not exceed $R_{sap}$.

## 5. Hardware-in-the-Loop Simulation and Results

A hardware-in-the-loop simulation system was established in order to verify the autonomous tracking method of Shenzhou reentry capsules based on video detection and heterogeneous UAV swarms. The video detection algorithm based on deep learning and the heterogeneous UAV swarm self-organizing control method based on tracking strategy

were verified. The results demonstrated the effectiveness of the autonomous tracking method proposed in this paper.

### 5.1. Hardware-in-the-Loop Simulation system

In order to verify the performance of the proposed method, the simulation system is mainly designed according to the following principles: (1) Test the performance of real-time video detection based on deep learning. (2) Test the performance of autonomous cooperative control of UAV swarms. (3) Test the correctness and compatibility of hardware and software configuration. (4) Test the exception-handling function.

The designed hardware-in-the-loop simulation system is shown in Figure 11. The red part on the left contains four autonomous intelligent controller nodes to simulate the onboard intelligent processor of four UAVs. This node integrates the intelligent video detection algorithm and the UAV swarm autonomous cooperative flight control method proposed in this paper. According to the input reentry capsule forecast data and the real-time videos acquired by the UAVs, the nodes perform self-organizing planning control of UAV swarms to achieve autonomous tracking of the reentry capsule. The middle blue part is the UAV swarm flight dynamics system, which is used to simulate UAV flight and ground control. It is composed of a flight control ground station, UAV dynamics simulation, graphics visualization workstation, HD display, map server, etc. It is the central node of the whole system and can display all UAV cluster control information, spacecraft reentry information and situation image information, etc. The green part on the right is the landing aera generator, including landing point forecast module, tracking strategy module, etc.

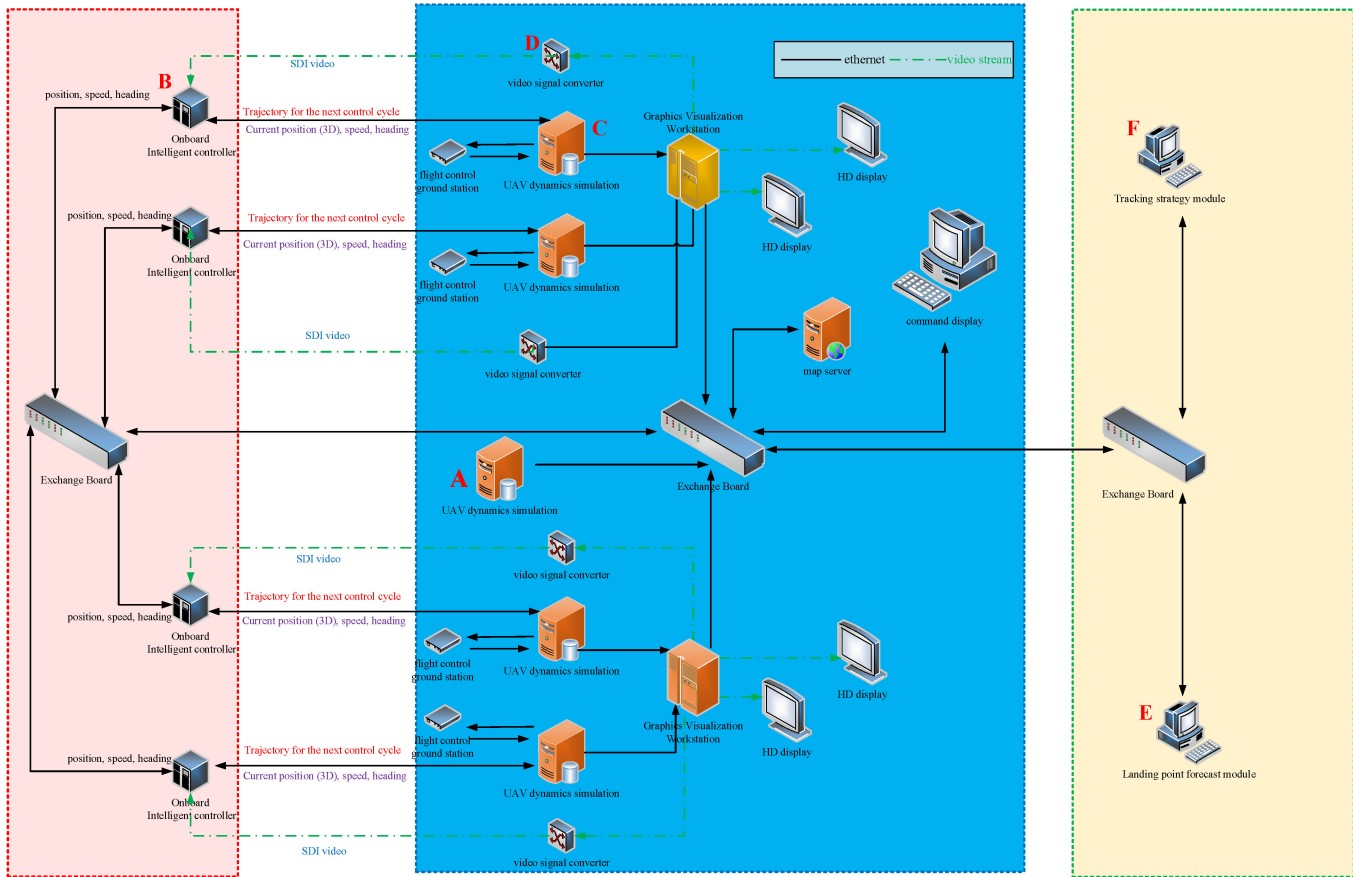

**Figure 11.** Hardware-in-the-loop simulation system.

In the initial stage of the simulation, the autonomous intelligent controller nodes will control the UAV swarm flight according to the predicted landing point trajectory. When the reentry capsule appears in the real-time video stream obtained by the UAV optical

pod, the autonomous intelligent controller will automatically detect the reentry capsule and calculate the real-time 3D position. Then, the UAV swarm will carry out coordinated flight based on three designed strategy modes to keep autonomous tracking of the reentry capsule. After the reentry capsule lands, the UAV swarm will complete the detection and tracking tasks. The UAV swarm will be recalled according to the aggregation mode, and the operation of the entire simulation system will be completed. The video detection and location data will be recorded in real time.

### 5.1.1. Detection Accuracy Test and Analysis

The purpose of the test is to analyze the accuracy of the video detection algorithm of Shenzhou reentry capsule in different states. A video of the capsule with parachute and a video of a single-capsule body are selected as detection data to test the accuracy of the proposed video detection algorithm. The data comes from the real landing video of the Shenzhou reentry capsule.

The result of the single-capsule state test is shown in Figure 12. The single-capsule state detection accuracy rate (mAP0.5) is 89.8%.

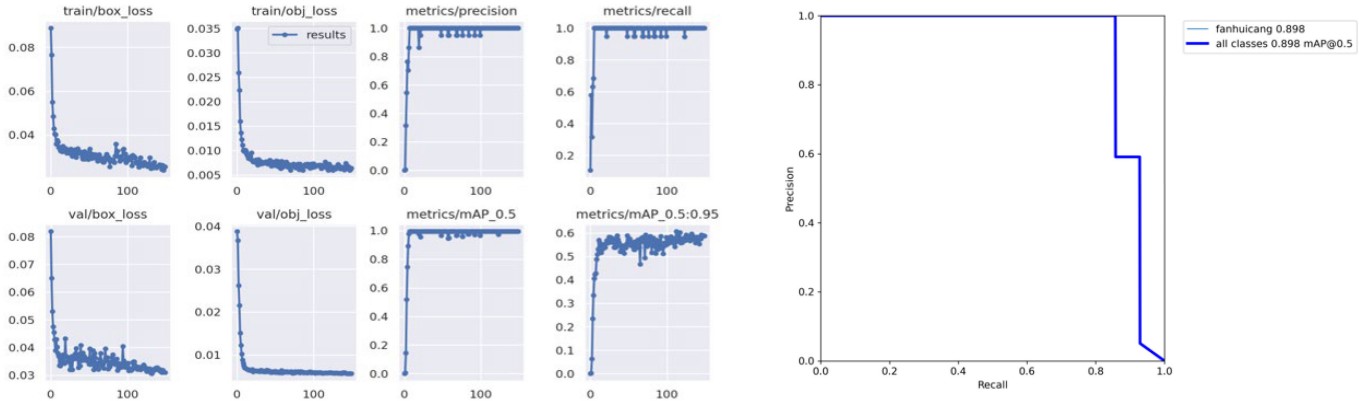

**Figure 12.** The result of the single capsule.

The result of the capsule with parachute test is shown in Figure 13. The detection accuracy rate (mAP0.5) of the capsule with parachute is 99.5%.

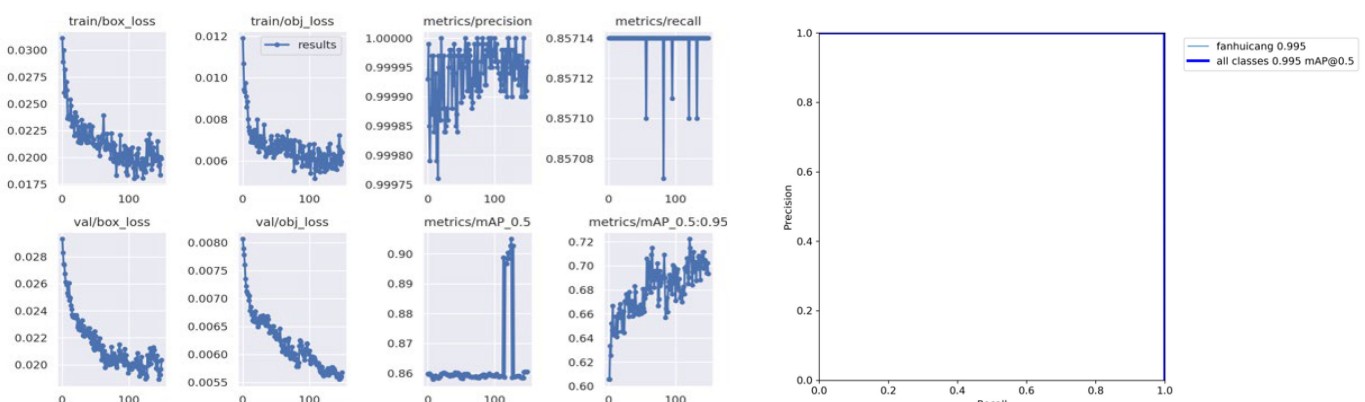

**Figure 13.** The result of the capsule with parachute.

### 5.1.2. Detection Frame Rate and Tracking Reliability

The detection time and the number of frames lost in recognition are tested and counted, as shown in Figure 14. Tracking reliability is 98.9%, and the detection frame rate is more than 135 Hz. Tracking reliability is defined as the ratio of the number of frames in which the target is recognized to the total number of frames.

```
Run:     detect_cabin ×
  ▶  ↑   video 1/1 (2994/3002) F:\Data\test3.mp4: 384x640 1 fanhuicang, Done. (0.005s)
  ■  ↓   video 1/1 (2995/3002) F:\Data\test3.mp4: 384x640 1 fanhuicang, Done. (0.005s)
         video 1/1 (2996/3002) F:\Data\test3.mp4: 384x640 1 fanhuicang, Done. (0.005s)
         video 1/1 (2997/3002) F:\Data\test3.mp4: 384x640 1 fanhuicang, Done. (0.005s)
         video 1/1 (2998/3002) F:\Data\test3.mp4: 384x640 1 fanhuicang, Done. (0.004s)
         video 1/1 (2999/3002) F:\Data\test3.mp4: 384x640 1 fanhuicang, Done. (0.005s)
         video 1/1 (3000/3002) F:\Data\test3.mp4: 384x640 1 fanhuicang, Done. (0.005s)
         video 1/1 (3001/3002) F:\Data\test3.mp4: 384x640 1 fanhuicang, Done. (0.005s)
         video 1/1 (3002/3002) F:\Data\test3.mp4: 384x640 1 fanhuicang, Done. (0.005s)
         Speed: 0.7ms pre-process, 5.3ms inference, 1.3ms NMS per image at shape (1, 3, 640, 640)
         detection rate is0.989, frame rate is 204.0
         Results saved to runs\detect\exp29

         Process finished with exit code 0
```

**Figure 14.** Tracking reliability and detection frame rate.

### 5.1.3. Target Capture Size and Robustness

The purpose of this test is to detect the smallest reentry capsule size that can be captured on video. The smallest size of the reentry capsule that can be detected is $9 \times 9$ pixels, as shown in Figure 15a. The detection performance with interference objects is shown in Figure 15b, which proves that the proposed algorithm has good robustness.

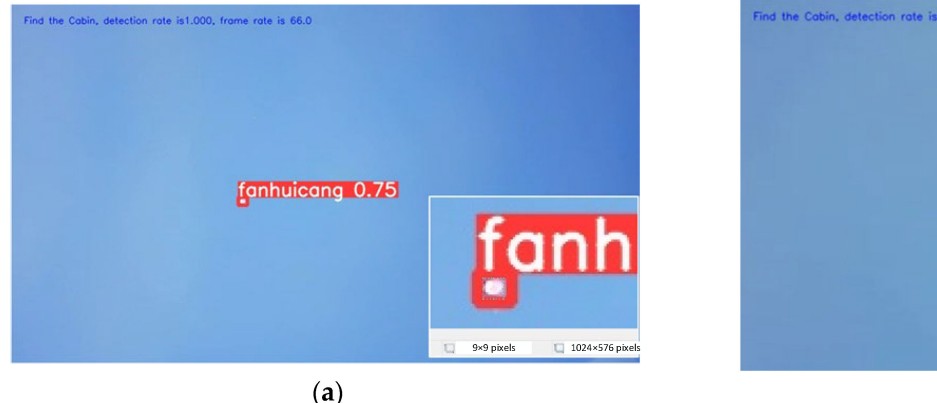
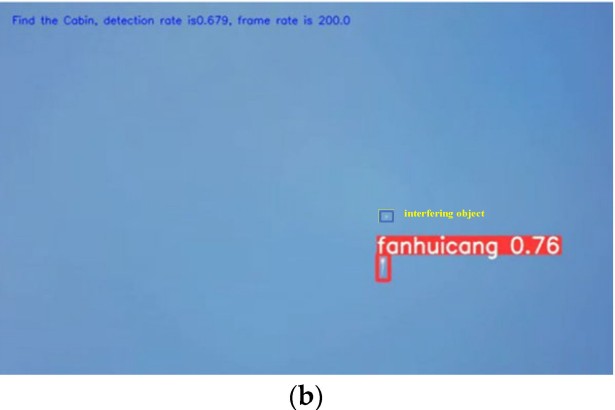

(**a**)                 (**b**)

**Figure 15.** Target capture size and robustness. (**a**) The smallest size. (**b**) Detection with interference.

### 5.1.4. Reentry Capsules Recognition in Different States

The ability to recognize the reentry capsule in different states was tested in our paper. We tested the detection performance of the return capsule in four different states, including touching moment, descending with the parachute, reverse rocket ignition state, and final landing state, as shown in Figure 16.

The above results show that the proposed video detection algorithm has good performance and is suitable for the Shenzhou reentry capsule autonomous tracking mission.

### 5.2. Test for Cooperative Flight Control Method of UAV Swarms

The numerical simulation test of the proposed cooperative flight control method of UAV swarms was carried out. The kinematic parameters of the heterogeneous UAV swarm are set according to Table 1. The number of fixed-wing UAVs is 4, and the number of rotary-wing UAVs is 50.

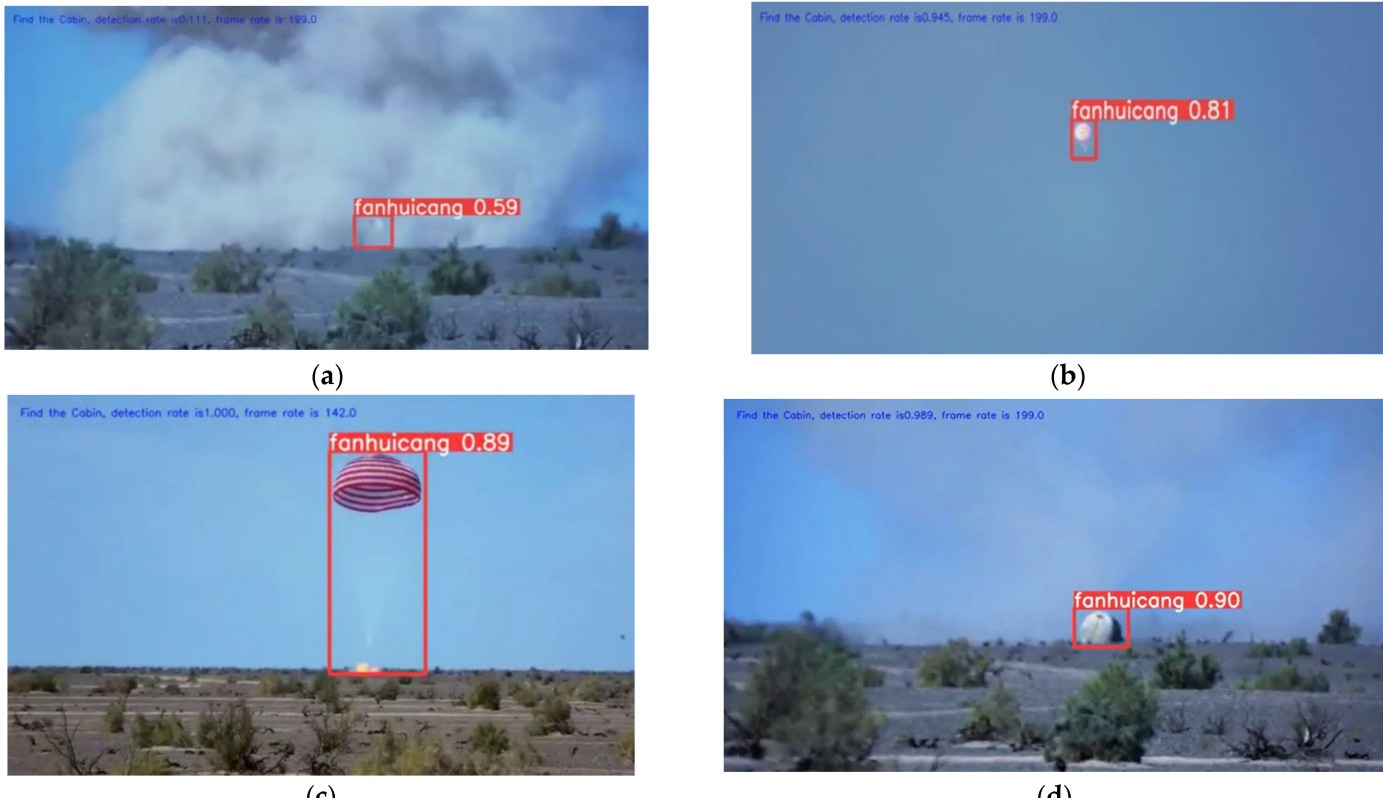

**Figure 16.** Reentry capsules recognition in different states. (**a**) Transient state of touchdown. (**b**) Descending with the parachute. (**c**) Reverse rocket ignition state. (**d**) Final landing state.

5.2.1. Initial Configuration and Cruise Mode Test

The fixed-wing UAVs take off from the airport uniformly and fly to the standby area under the control of the virtual potential field. Rotary-wing UAV are initially distributed near the predicted landing point and complete the initial configuration under the action of control force. The UAV swarm initial configuration and cruise process are shown in Figure 17. The blue line represents the landing trajectory of the reentry capsule. The red triangles represent the fixed-wing UAVs. The blue dots represent the rotary-wing UAV. The configuration of the UAV swarm is shown in the Figure 17f.

5.2.2. Relay Tracking and Observation Mode Test

Under this strategy mode, the fixed-wing UAVs will autonomously track and detect the reentry capsule in sequence. Then relay tracking is performed by the rotary-wing UAVs. The process of the fixed-wing UAVs tracking is shown in Figure 18.

The process of the rotary-wing UAV tracking is shown in Figure 19. The green oval represents the safe radius of reentry capsule. The blue circle represents the relay tracking altitude. The red points represent the rotary-wing UAVs that can capture the reentry capsule. After the capsule lands, the rotary-wing UAVs perform a fly-around observation of the reentry capsule at the configuration altitude, as shown in Figure 19c,d.

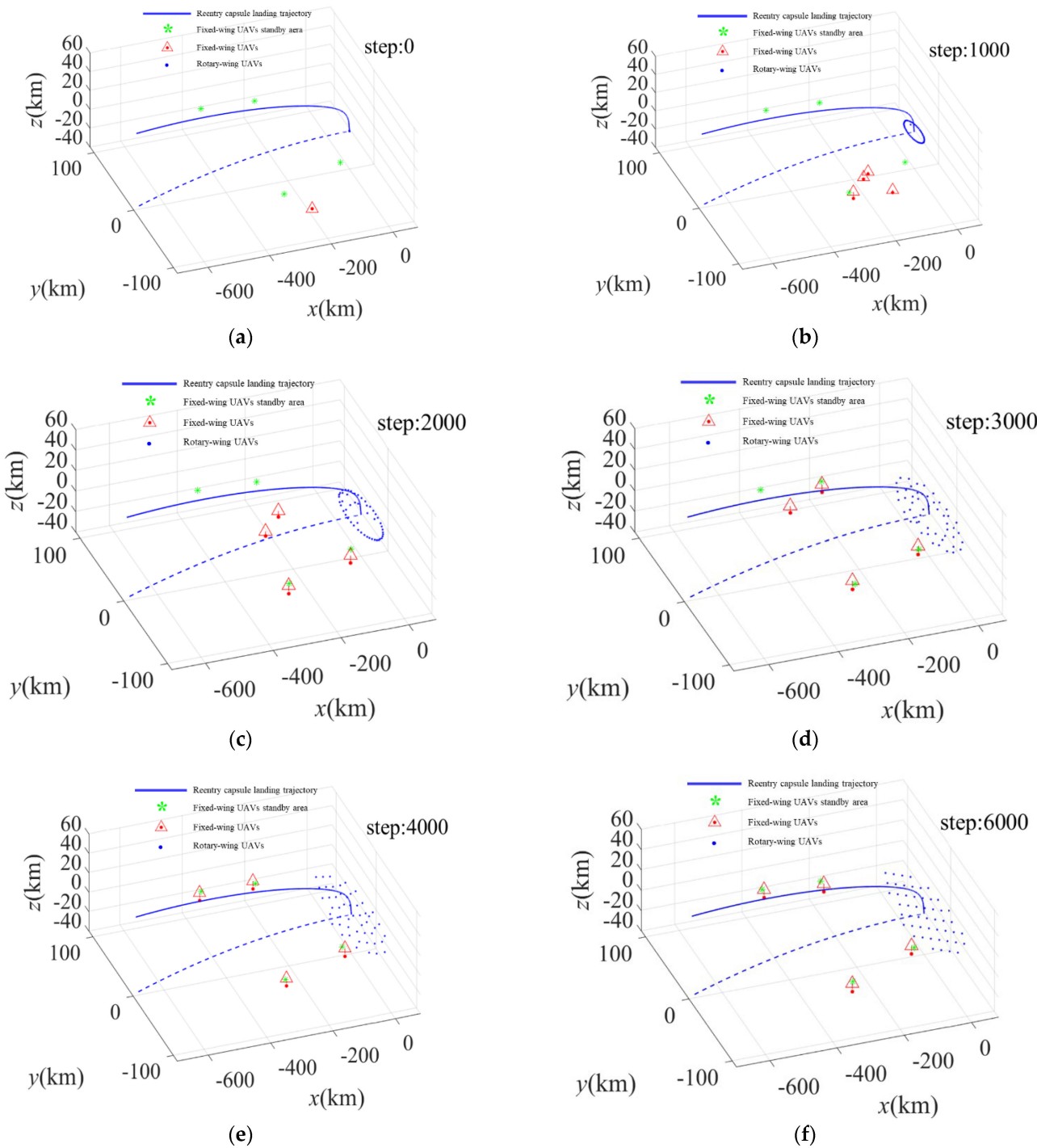

**Figure 17.** Initial configuration and cruise mode test. (**a**–**f**) are the results of several step of numerical simulation.

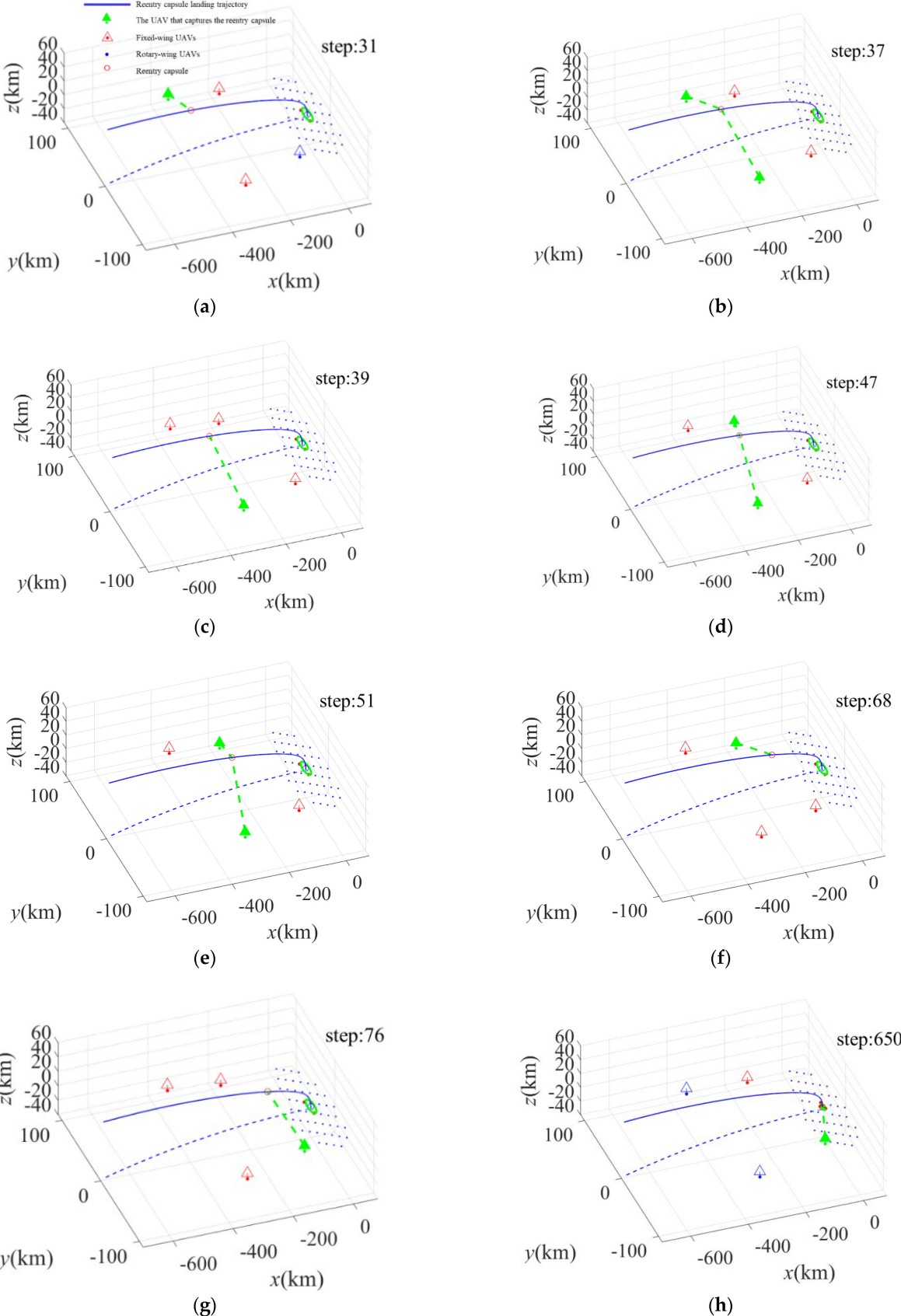

**Figure 18.** The process of the fixed-wing UAVs tracking. (**a**–**h**) are the results of several step of numerical simulation.

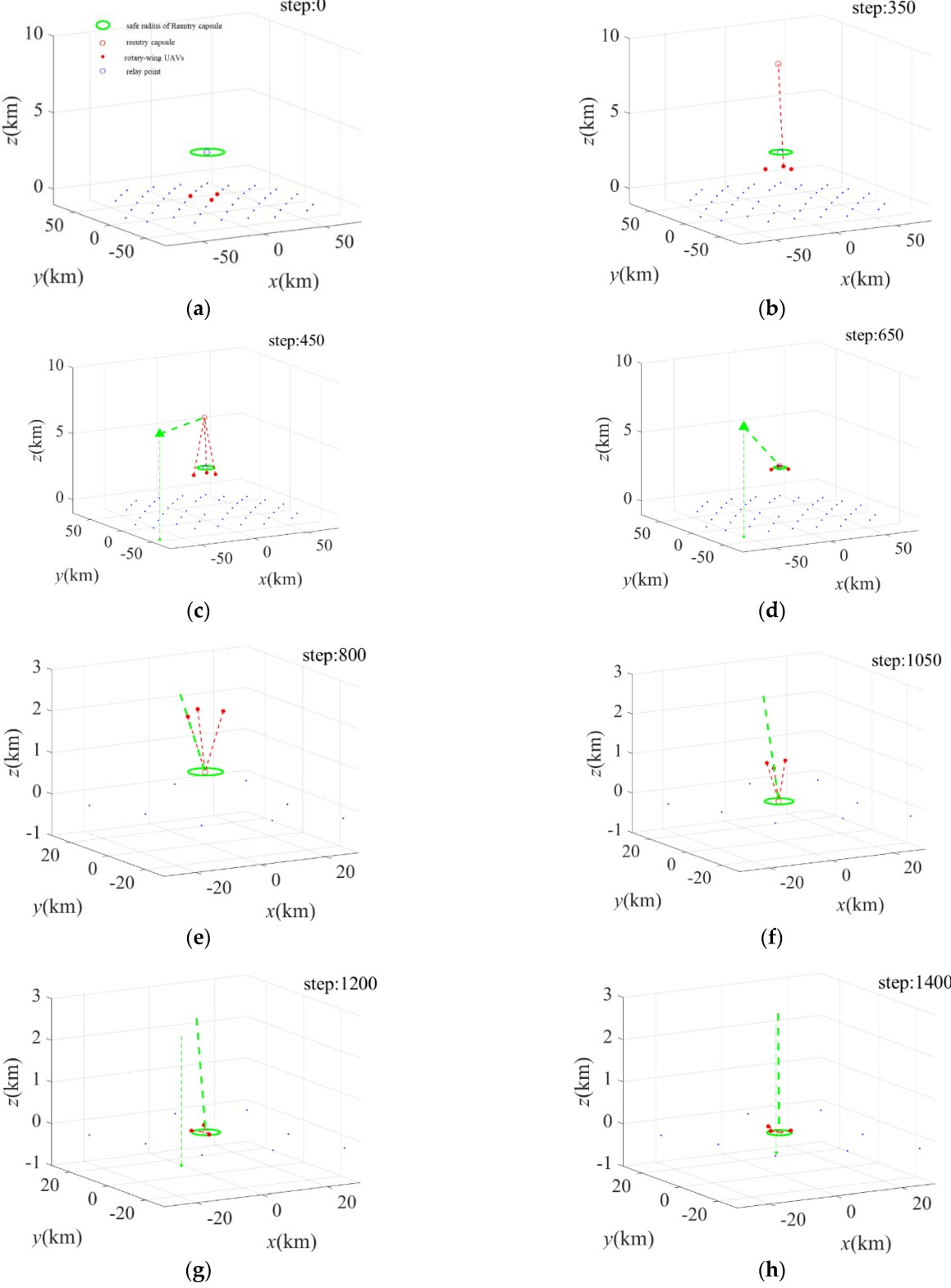

**Figure 19.** The process of the rotary-wing UAVs tracking. (**a**–**h**) are the results of several step of numerical simulation.

### 5.2.3. Aggregation Mode Test

After completing the tracking and recovery of the return capsule, the UAV swarm will be recalled according to the aggregation mode. The process of UAV swarm aggregation is shown in Figure 20.

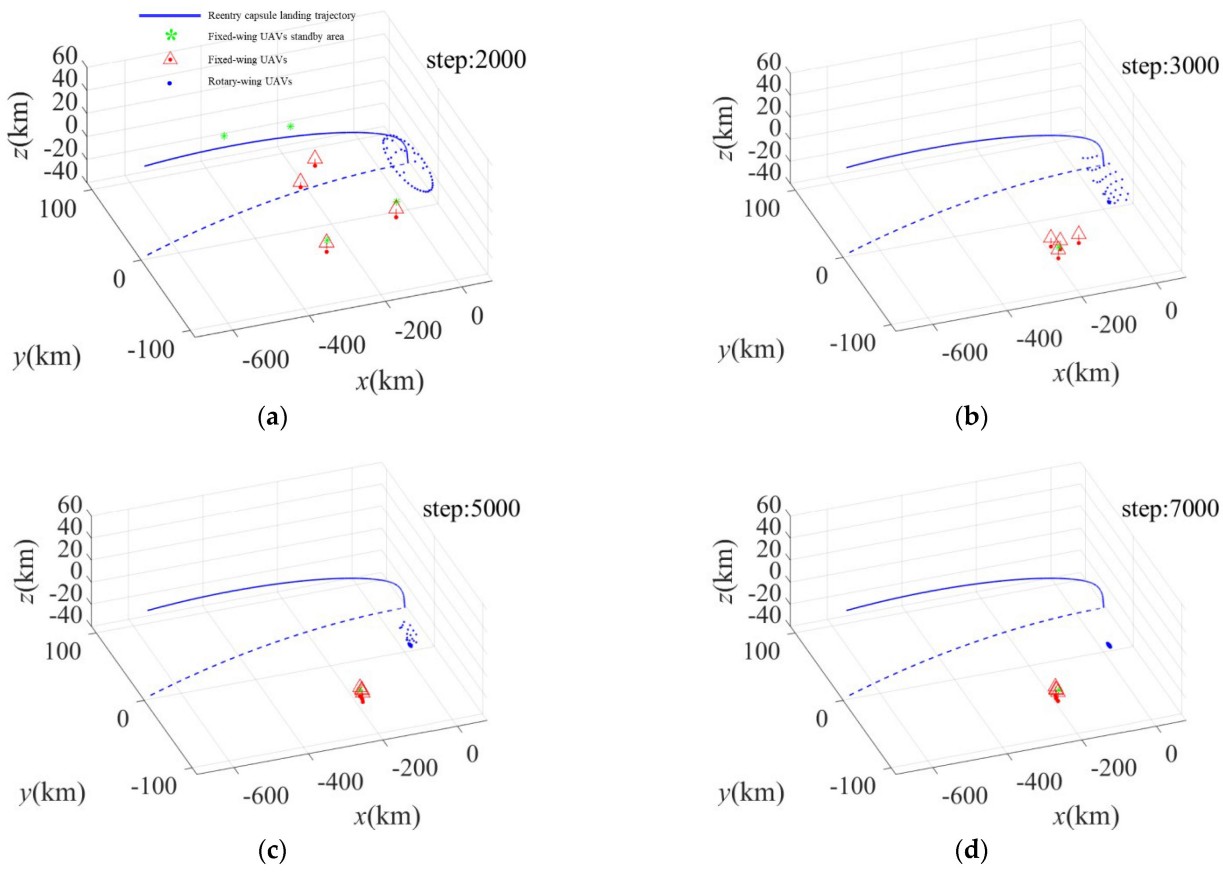

**Figure 20.** Aggregation mode test. (**a**–**d**) are the results of several step of numerical simulation.

### 5.3. Test for Autonomous Tracking of Shenzhou Reentry Capsules

After the individual test and verification of the video detection algorithm and the cooperative flight control method, the whole process verification of the autonomous tracking for Shenzhou reentry capsule was carried out by the hardware-in-the-loop simulation system. The test process and specific scenarios are shown in Figure 21.

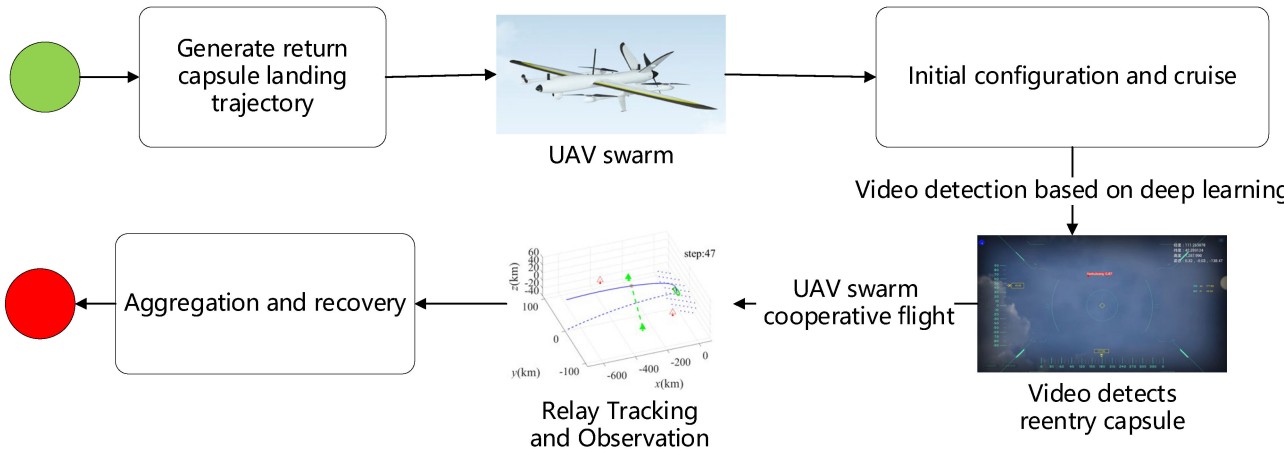

**Figure 21.** The test process.

The simulated system can be completely replicated the real world, as it contains all the parameters of the actual control of the UAV swarm. The entire simulation process and the video from the UAV optical pod will be visualized in real time.

### 5.3.1. Shenzhou Reentry Capsule Detection and Visual Lock

When the UAV captures the Shenzhou reentry capsule, the optical pod will be locked and focused on the capsule, as shown in Figure 22. The specific position of the reentry capsule will be calculated based on the results of video detection and the pose of UAV.

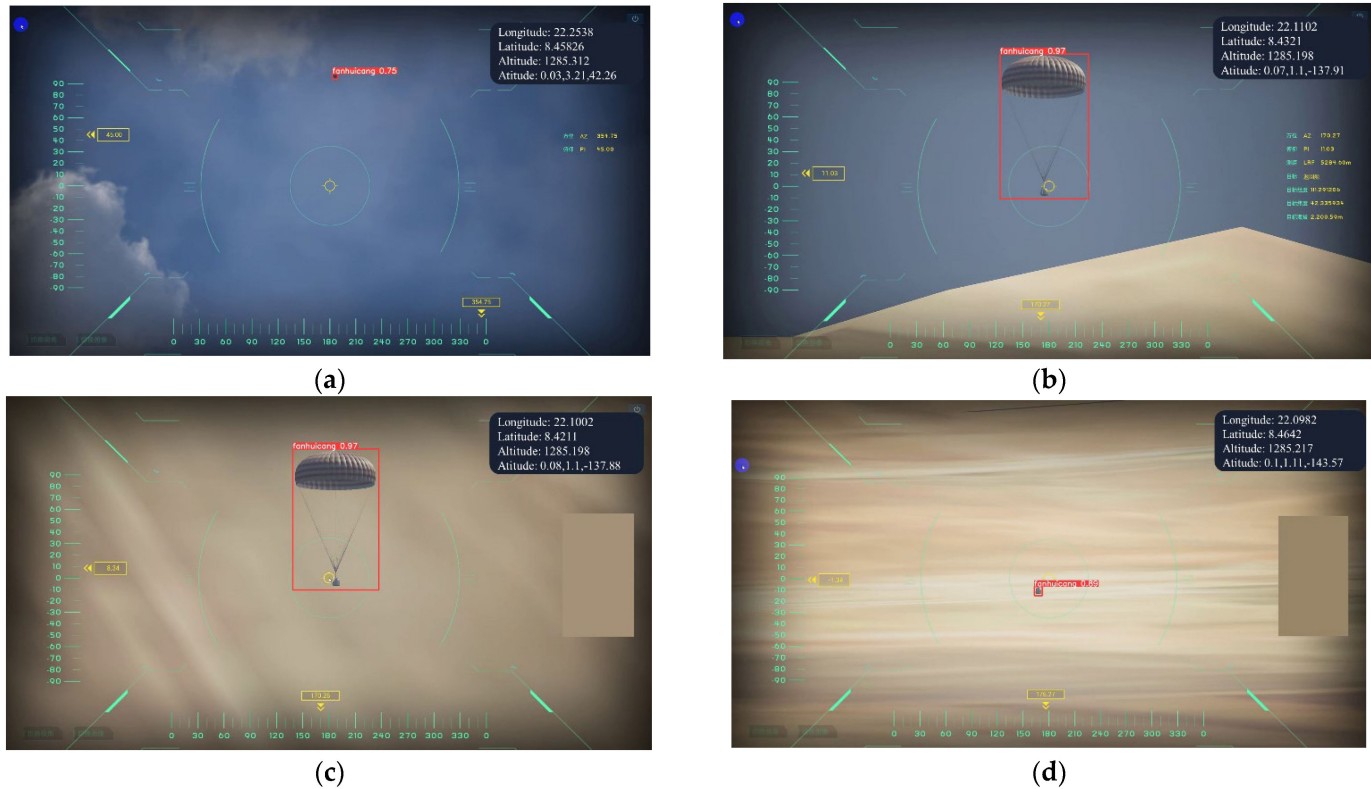

**Figure 22.** Shenzhou reentry capsule detection. (**a**–**d**) represent the process from capture to continuous tracking of the return capsule.

### 5.3.2. Autonomous Tracking of Shenzhou Reentry Capsules

After obtaining the position of Shenzhou reentry capsule, the UAV swarm performs cooperative flight and autonomous tracking according to the proposed strategy. The tracking trajectory of each UAV is shown in Figure 23. The red line represents the landing trajectory of the Shenzhou reentry capsule. Each UAV detects the Shenzhou reentry capsule from different standby area. After the reentry capsule is captured, the UAV performs relay tracking for the capsule at different altitudes.

The above results show that the method proposed in this paper to autonomously track the Shenzhou reentry capsule is effective and has good performance.

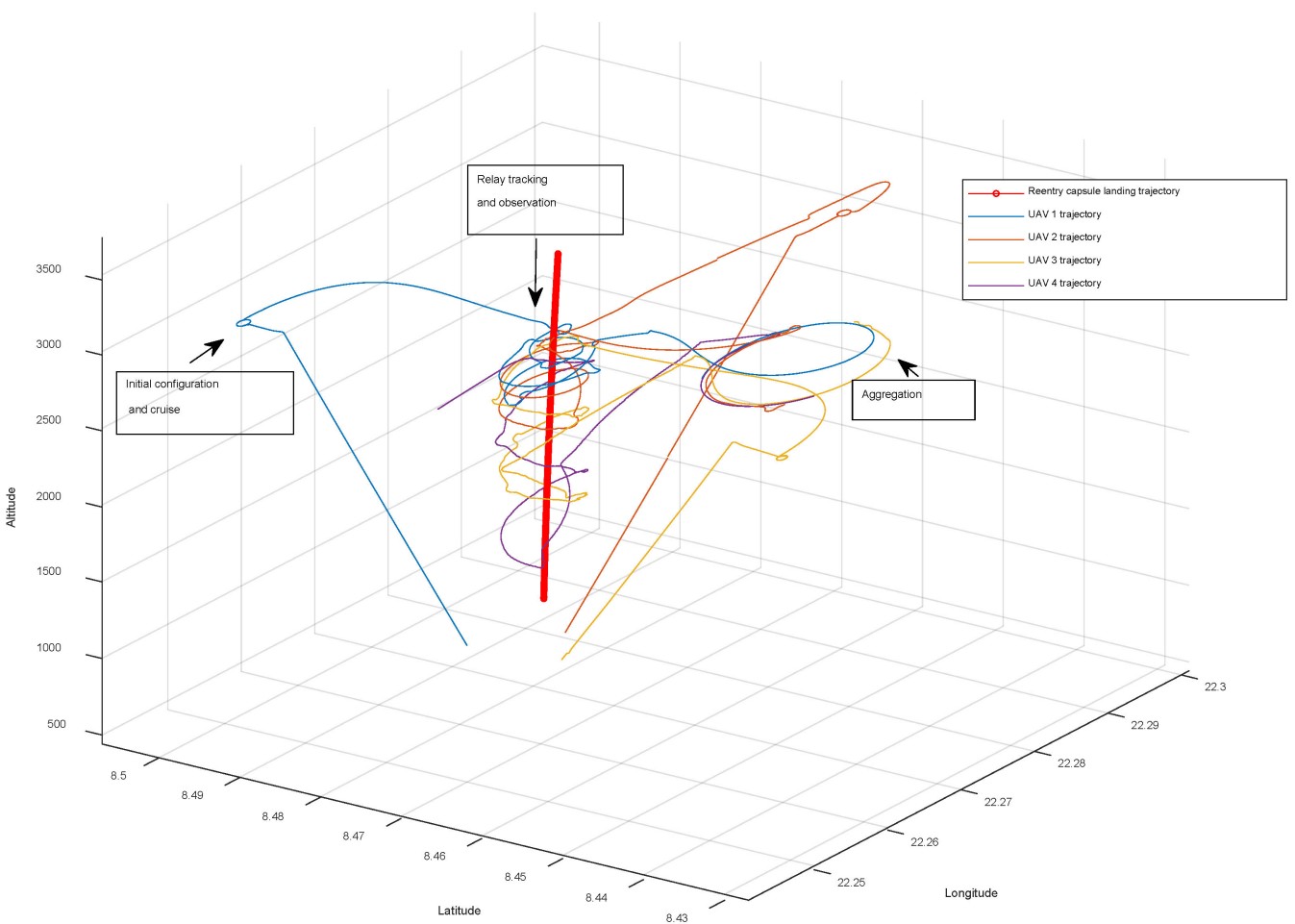

**Figure 23.** Tracking trajectory of each UAV.

### 6. Conclusions

The safe landing and rapid recovery of the reentry capsules are very important to the manned spacecraft missions. Aiming at the challenge of Shenzhou reentry capsules tracking and observation, the paper proposes a new approach for autonomous tracking based on video detection and heterogeneous UAV swarms. An autonomous tracking strategy is designed to satisfy the different states of reentry capsules. A multi-scale video target detection algorithm based on deep learning is developed to recognize Shenzhou reentry capsules and obtain positioning data. Additionally, we proposed a self-organizing control method based on virtual potential field for the cooperative flight of UAV swarms.

In order to verify the performance of the autonomous tracking method, we establish a hardware-in-the-loop simulation system. The test covers the video detection algorithm, the UAVs cooperative flight control method and the whole process of autonomous tracking. The results show that the reentry capsule can be detected in least four different states, and the detection accuracy rate of the capsule with parachute is 99.5%. The proposed autonomous tracking method can effectively control the UAV swarm to track the Shenzhou reentry capsule based on the video intelligent detection information. It is of great significance to the real-time searching of reentry capsules and guaranteeing astronauts' safety. The proposed autonomous tracking method for the Shenzhou reentry capsule has reference significance for future reentry capsule search and rescue.

**Author Contributions:** Conceptualization, B.L. (Boxin Li) and Z.W.; methodology, B.L. (Boxin Li) and B.L. (Boyang Liu); software, B.L. (Boxin Li); validation, B.L. (Boxin Li) and D.H.; formal analysis, B.L. (Boxin Li); investigation, B.L. (Boxin Li); resources, Z.W.; data curation, B.L. (Boxin Li); writing—original draft preparation, B.L. (Boxin Li); writing—review and editing, Z.W.; visualization, B.L. (Boxin Li); supervision, Z.W.; project administration, Z.W.; funding acquisition, Z.W. All authors have read and agreed to the published version of the manuscript.

**Funding:** This research was funded by the National Natural Science Foundation of China (No. 11872034).

**Data Availability Statement:** Data from this study are available from the corresponding author upon request.

**Acknowledgments:** This research was supported by the National Natural Science Foundation of China No. 11872034).

**Conflicts of Interest:** The authors declare no conflict of interest.

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
