# Peer review of "Autonomous Tracking of ShenZhou Reentry Capsules Based on Heterogeneous UAV Swarms"

_drones, doi:10.3390/drones7010020_

Round 1
Reviewer 1 Report
1. This research paper contains an impressive amount of detail and presents a comprehensive overview of autonomous tracking of ShenZhou reentry capsules based on video detection and heterogeneous UAV swarms.
2. The title of the paper should be revised for clarity and sequence.
3. The abstract should include more detail about the specific parameters of the mission.
4. The introduction is well written and comprehensive, though references from more recent studies could be added.
5. It should be made clear whether the heterogeneous UAVs are composed of two specific types or of subtypes.
6. Table 1 should include an abbreviation at least once before it is used frequently.
7. The numerical model should be presented in a clearer and more concise way and equations should be cited in the text.
8. The results section is well composed and logically presented.
9. The conclusion is logical and does a good job of answering the research question.
10. All in all, this research paper is an impressive and comprehensive study of autonomous tracking of ShenZhou reentry capsules based on video detection and heterogeneous UAV swarms.
Reviewer 2 Report
The paper address autonomous tracking of ShenZhou Reentry capsules based on video detection and heterogeneous UAV swarms.
General observations and formulation:
1. Figure 2 – for increasing the clarity it is recommended that the subtitles “Twin-Tailed Scorpion” and “X-Swift” to have a bullet letter (ex. a), b))
2. Some figures referred to in the text are bold and others are not, such as figure 4 - is there a specific reason?
3. Figures 16, 17, 18 starts on one page and continues the next page and create confusion because the name of the figure can only be seen on the second page.
4. In figure 21, various fonts have been used which create confusion - it is recommended to use only one font.
Substantive remarks:
1. A very well structured and written article.
2. In search and tracking tasks bad weather conditions, limited or absence of network connectivity, limited visual range, spread of the search zone, absence of GPS signal and other similar problems, are complex and detrimental to the use of UAV swarms - how do you relate to these challenges and under what conditions can they be kept under control through the solution proposed in this article.
3. Unmanned aerial vehicle (UAV) networks have a wide range of applications, such as Internet of Things (IoT), 5G communications, and so forth. However, the communications between UAVs and UAVs to ground control stations mainly use radio channels, and therefore these communications are vulnerable to cyberattacks - how do you relate to these challenges and under what conditions can they be kept under control through the solution proposed in this article.
4. Regarding the sustainability of such a system in terms of energy consumption, radio/ electromagnetic interference, physical obstacles, electrical shielding – please give some details.
5. In calamity situation 5G networks are not working. How does the system handle this situation.
6. How do you deal with the time delay issue of the system? – please address this
7. Please briefly detail in the conclusions what does it add to the subject area compared with other published material?
